# Barriers and Enabling Factors for Central and Household Level Water Treatment in a Refugee Setting: A Mixed-Method Study among Rohingyas in Cox's Bazar, Bangladesh

**Mahbub-Ul Alam [1,*]** , **Leanne Unicomb [1]** , **S.M. Monirul Ahasan [1,2]** , **Nuhu Amin [1]** , **Debashish Biswas [1]** , **Sharika Ferdous [1]** , **Ayesha Afrin [1]** , **Supta Sarker [1]** and **Mahbubur Rahman [1]**

[1]    International Centre for Diarrhoeal Disease Research, Environmental Interventions Unit, Infectious Diseases Division, Bangladesh, Dhaka 1212, Bangladesh; leanne@icddrb.org (L.U.); rajibanth4@gmail.com (S.M.M.A.); nuhu.amin@icddrb.org (N.A.); debashish@icddrb.org (D.B.); sharika.ferdous@icddrb.org (S.F.); ayesha.afrin@icddrb.org (A.A.); supta@icddrb.org (S.S.); mahbubr@icddrb.org (M.R.)
[2]    International Rescue Committee, Dhaka 1212, Bangladesh
*    Correspondence: mahbubalam@icddrb.org

**Abstract:** Water chlorination is widely used in emergency responses to reduce diarrheal diseases, although communities with no prior exposure to chlorinated drinking water can have low acceptability.   To better inform water treatment interventions, the study explored acceptability, barriers, and motivating-factors of a piped water chlorination program, and household level chlorine-tablet distribution, in place for four months in Rohingya refugee camps, Cox's Bazar, Bangladesh.   We collected data from June to August 2018 from four purposively selected refugee camps using structured observation, key-informant-interviews, transect-walks,  group discussions,  focus-group  discussions,  and  in-depth-interviews  with males, females, adolescent girls, and community leaders.   Smell and taste of chlorinated water were commonly reported barriers among the population that had previously consumed groundwater. Poor quality source-water and suboptimal resultant treated-water, and long-queues for water collection were common complaints.  Chlorine-tablet users reported inadequate and interrupted tablet supply, and inconsistent information delivered by different organisations caused confusion. Respondents reported fear of adverse-effects of "chemicals/medicine" used to treat water, especially fear of religious conversion. Water treatment options were reported as easy-to-use, and perceived health-benefits were motivating-factors.   In vulnerable refugee communities, community and religious-leaders can formulate and deliver messages to address water taste and smell, instil trust, allay fears, and address rumours/misinformation to maximise early uptake.

**Keywords:**   water treatment;  point-of-delivery;  point-of use (POU); barriers of water treatment; enabling factors for water treatment; Rohingya refugees; qualitative; mixed-methods; Cox's Bazar; Bangladesh

## 1. Background

Amidst the myriad of health issues for which Rohingya refugees in Bangladesh are at risk, waterborne illness is one of the most common [1,2]. Contaminated water, lack of safely managed sanitation, and inadequate hygiene facilities are a concern for diarrheal disease transmission among the community [3,4]. In this emergency context, the provision of safe drinking water is crucial to reduce and control diarrhoea and other waterborne diseases [5]. Within the new settlements for Rohingya that

emerged after August 2017, there were no pre-existing water, sanitation, and hygiene (WASH) facilities including toilets, water points, or bathing places, and some people reported collecting water from the paddy fields for drinking [6]. Water points were installed in the Rohingya camps, but the water was not safe to drink; approximately 28% of the source water was contaminated with faecal coliforms, and 10.5% with *E. coli* among 3186 tubewells tested [1]. The early stages of this emergency were tackled by humanitarian actors stepping up to ensure basic WASH infrastructure, where quantity was prioritised but facilities were temporary in nature, often overlooking quality in terms of water point depth, location, and functionality. Ensuring safe drinking water management at the household level remains challenging, and household level water contamination remains high [1].

Approximately 742,000 Rohingya fled to Cox's Bazar in Bangladesh from 25 August 2017 due to extreme violence and persecution in the western Rakhine State of Myanmar [7]. In Myanmar, almost all of them used untreated groundwater (tubewell water) for drinking [8]. The Rohingya populations, mostly Muslim, were rehabilitated in 30 newly built camps (around 30,000 per camp), in addition to two pre-existing registered camps in the Ukhiya and Teknaf sub-districts of Cox's Bazar. Although the majority of households in all refugee camps collected water exclusively from tubewells, a small number collected drinking water from rivers and other unspecified sources [9].

Chlorine-based water treatment is a rapid and cost-effective water decontamination method [10]. However, it is difficult to effectively implement chlorination programs in emergencies, since research on user acceptability of various chlorination methods from emergency contexts is scarce [8,10]. Furthermore, there are multiple standard recommendations for the amount of chlorine (dosing) needed to adequately treat water or the level of residual chlorine that should remain in the water after initial treatment [8,10,11]. Water chemistry may vary geographically, which also impacts chlorine demand and therefore dosing. In addition, chlorine acceptance is not universal and—as found from previous studies—many who have treated water with chlorine products have disliked the additional burden of treating their household water supply, and the taste and odour thresholds may vary [10,12].

A chlorinated water supply has been commonly used in high-income countries and is known as an effective and reliable method to treat drinking water [13]. However, there have been no evaluations of piped water chlorination in systematic reviews of interventions implemented in humanitarian responses [14]. Some evidence exists that the user's acceptance is higher in emergencies due to an increased level of fear of disease risk. However, recent research on chlorination in emergencies concludes that there is a genuine lack of information on user acceptability of chlorination in emergencies [15]. It is essential to acknowledge that programmatic elements cannot necessarily be generalised for different types of emergencies or geographic and cultural settings.

This study aimed to inform future water treatment interventions in emergencies and was conducted in response to low uptake of two chlorination methods, one at the point of delivery and another at use. Uptake of safe drinking water to reduce waterborne disease in these communities increased ultimately after the initial settlement of the populations. We explored the perceptions and acceptability of ongoing water treatment programs, barriers, and motivating factors for water treatment for Rohingya refugee communities in Cox's Bazar, Bangladesh.

## 2. Methods

### 2.1. Study Area

We conducted a mixed-method study in four out of 30 Rohingya refugee camps in Ukhiya and Teknaf Upazilas (sub-districts), Cox's Bazar, Bangladesh (Figure 1) between June and August 2018. The study explored two water treatment programs out of three used by populations in Rohingya camps: water distribution networks with piped water chlorination (Camp 22, Camp 26), and household point of use (POU) water treatment using chlorine tablets (Aquatab®, Medentech, Wexford, Ireland)

distributed by NGOs (Camp 4, and Camp 18). Study sites were selected purposively from 15 camps where Oxfam had been implementing WASH programs.

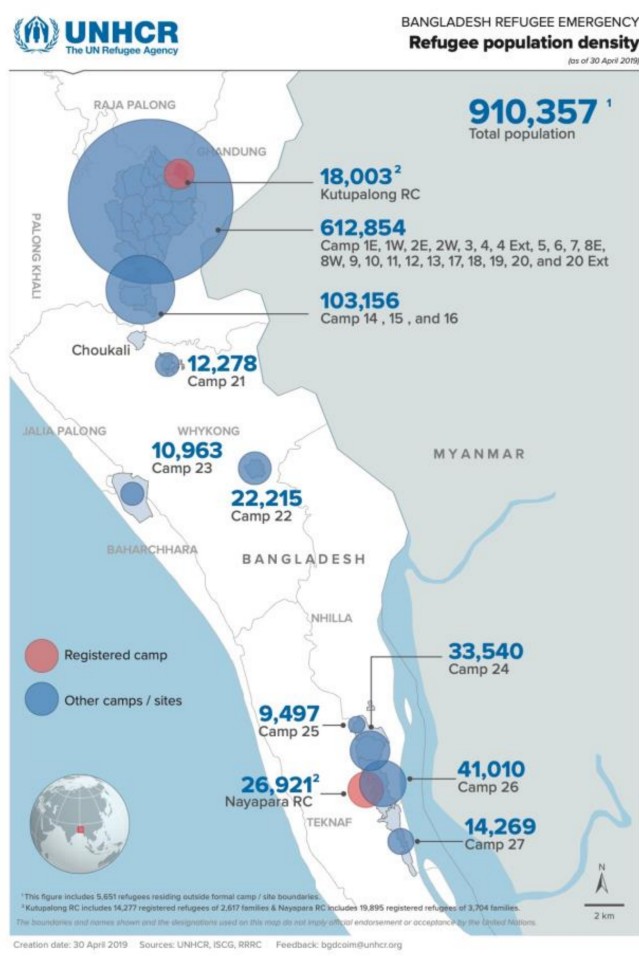

**Figure 1.** Rohingya camps in Cox's Bazar (Source: UNHCR); study sites camps 4, 18, 22, 26.

*2.2. Water Treatment Program Delivery and Communication with Community Members*

Several chlorination programs were implemented to ensure the provision of safe water, varying according to the WASH implementing partner and area: (a) point of delivery (in-line chlorination and piped water chlorination), (b) point of source (bucket chlorination), and (c) POU (Aquatab distribution) [16]. Oxfam implemented the POU intervention from the early response (from October 2017) and started point of delivery intervention subsequently (from April 2018) [16].

In Camp 22 and Camp 26, water from a surface water treatment plant was provided from an established supply network. In Camp 4 and Camp 18, inhabitants used various water sources including shallow tubewells, deep tubewells, or a small stream.

The international and local program staff, who were not from the Rohingya community, developed the program messages, and beneficiaries varied between programs. Household POU water treatment using chlorine tablets required volunteers who interacted with beneficiaries to deliver chlorine tablets and behaviour change communication (BCC). Volunteers (who were from the Rohingya community and usually community leaders (*Majhis*) were asked by Oxfam to deliver messages to the beneficiaries pertaining to drinking treated water, the need to wait 30 min from treatment until use, and the need to collect fresh water each day. This information was shared by volunteers during meetings, by megaphones, or door to door visits. Piped water chlorination staff (who were local staff but not from the Rohingya community) did not have any direct interaction with beneficiaries;

however, staff separately implementing camp-wide hygiene promotion activities promoted the drinking of chlorinated water [16].

*2.3. Data Collection*

We used quantitative and a range of qualitative data collection techniques to achieve our objectives. The qualitative techniques included key-informant-interviews (KII), transect-walks, group discussions (GD), focus group discussions (FGD), and In-depth interviews (IDI). The quantitative component included structured observation and drinking water sample testing for free and total chlorine and turbidity.

A team of anthropologists including four males and four females with training in qualitative research were responsible for collecting the data. The team conducted four transect walks to obtain an understanding of the overall camp context and to explore the location of water points and the drinking water supply system in study areas.

Volunteers from the Rohingya community and two researchers participated in transect walks and developed illustrated maps for each study site indicating the water points, toilets, and public places such as bazaars, education centres, mosques, medical centres, and child safety zones. The research team developed and used a checklist, which included the items to observe while talking to participants during the walk. After finishing the walk, the team sat with transect walk participants and provided them with large sheets of paper and coloured pens to draw maps of their locality. Data from the transect walk was used to refine the structured observation checklist, the IDI, and FGD data collection checklists.

The team conducted eight group discussions (four with adult males and four with adolescent boys) to explore their contribution to different aspects of water management and water treatment in the community (i.e., water collection, treatment at home, treatment product collection). In-depth interviews were conducted with eight adult women, four adolescent girls, and four men to understand their water collection, storage, and water treatment practices, and perceptions related to water quality. In-depth interview participants were purposively selected from different parts of each camp to maximise participant variability. FGDs included four sessions with adult females and four with adolescent girls, one in each of four camps, to explore their perceptions and practices related to water collection and water treatment, as studies suggested that they were predominantly responsible for water collection, treatment, and use [17]. The team conducted four KIIs with community leaders who were locally known as *Majhi* and four with influential females, known as *Hafeza*, to explore the community engagement during an intervention and their influence on the community. We chose respondents considering their influence on the community after an informal discussion with the community members. For all data collection techniques, the team used a separate interview guideline with open-ended questions covering a priori themes, most of which emerged during the discussion with transect walk participants. The study team members reviewed all guidelines before data collection.

The team conducted two hours of structured observations among 103 households in 26 compounds focusing on key water management events, including water collection, location of the water source, water collection time, water storage methods, water vessel type, water treatment, and water use among household members using a standard data collection instrument developed by water, sanitation, and hygiene experts at International Centre for Diarrhoeal Disease Research, Bangladesh (icddr,b). The team conducted structured observations in compounds comprised of 3–5 households per compound. The team selected households that were located within a 10 min walking distance from the water collection points to observe as many water-related activities as possible.

*2.4. Water Sample Collection and Analysis*

A trained laboratory assistant collected water samples from households and shared water points to measure free and total residual chlorine. The laboratory assistant also measured water turbidity as an indicator of chlorine demand. We collected water samples from 103 (structured observation)

households and 26 shared primary drinking water sources (tubewell, standpipe, and/or shared taps) adjacent to these households.

In the field, the team used a digital colourimeter (LaMotte Model 1200, LaMotte Company, Chestertown, MD, USA) to detect free and total chlorine, and a turbidity meter (LaMotte Model 2020i, LaMotte Company, Chestertown, MD, USA) to detect turbidity [17].

### 2.5. Data Analysis

The team captured audio recordings of interviews and group discussions. If study participants objected to an audio recording, the researchers took detailed notes and expanded the data in detail after returning from the field.

After each day of data collection, the field team shared their day's experience with other team members. The study coordinator and qualitative research supervisor developed a data entry matrix to capture the data. On the same or subsequent day, data collectors listened to audio recordings and entered relevant findings into the pre-developed data matrix using an MS Excel spreadsheet according to a priori codes. They incorporated relevant emergent codes and findings into the matrix. Members of the team checked the data matrix for reliability and consistency. They inserted all findings in Bengali into the matrix, and subsequently, the analysis was drafted in English.

For each water treatment method, we defined two user groups: doers and non-doers. During the interview, we ascertained respondent treated water use status, and during analysis, we used these to compare, explain, and interpret results.

- 'Doers'—those that had access to chlorine (Aquatabs) and used/chose piped water chlorination sources daily or most of the time
- 'Non-Doers'—those that had access to chlorine and used/chose piped water chlorination sources on an intermittent basis or chose not to use at all

We provided summary statistics on participant characteristics and for other key indicators. Structured observation data were analysed and presented by the number of events observed. We presented mean free and total chlorine in mg/L, and turbidity in Nephelometric Turbidity Unit (NTU).

### 2.6. Ethical Approval

The study protocol was approved by the Ethics Review Committee of the International Centre for Diarrhoeal Disease Research, Bangladesh (icddr,b). Study participants were informed of the aims of the study and their rights. Enumerators read an information sheet to respondents in the local language, answered any questions raised, and obtained written consent for participation. Respondents were given a copy of the information sheet to keep, and no compensation was provided for participation. Names and numbers were removed from final data sets to protect anonymity.

## 3. Results

### 3.1. Respondents' Characteristics

The median age of adult participants ranged from 34 to 35 years and adolescents from 14 to 16 years. Of the 230 participants, 52% (120) had no formal education (Table 1). Almost all of the respondents arrived between September 2017 and March 2018. Respondents reported that they were not involved in any income generation activity.

**Table 1.** Characteristics of study participants and water source used, by camp.

| Socio-Demographic Characteristics | Camp 4 | Camp 18 | Camp 22 | Camp 26 | Overall |
|---|---|---|---|---|---|
| Number of participants | 53 | 57 | 58 | 62 | 230 |
| Median age—Adult (in years) | 35 | 34 | 34 | 35 | 35 |
| Median age—Adolescents (in years) | 14 | 15 | 16 | 14 | 15 |
| Education: | n (%) | n (%) | n (%) | n (%) | n (%) |
| No formal education | 26 (49) | 28 (49) | 30 (52) | 36 (58) | 120 (52) |
| Grade 1 to 5 | 12 (23) | 14 (25) | 16 (28) | 18 (29) | 60 (26) |
| Grade 6 to 9 | 10 (19) | 11 (19) | 6 (10) | 6 (10) | 33 (14) |
| Grade 10 and above | 4 (8) | 3 (5) | 5 (9) | 1 (2) | 13 (6) |
| *Hafez* * | 1 (2) | 1 (2) | 1 (2) | 1 (2) | 4 (2) |
| Water sources: | n (%) | n (%) | n (%) | n (%) | n (%) |
| Shallow tubewell | 33 (85) | 22 (55) | 0 (0) | 5 (14) | 60 (38) |
| Deep tubewell | 6 (15) | 1 (2.5) | 0 (0) | 0 (0) | 7 (4.4) |
| Piped water chlorination | 0 (0) | 1 (2.5) | 19 (45) | 23 (62) | 43 (27) |
| Rain water | 0 (0) | 0 (0) | 7 (17) | 0 (0) | 7 (4.4) |
| Cisterns/well | 0 (0) | 0 (0) | 8 (19) | 0 (0) | 8 (5.1) |
| Small streams | 0 (0) | 3 (7.5) | 3 (7.1) | 9 (24) | 15 (9.5) |
| Puddles | 0 (0) | 3 (7.5) | 2 (4.8) | 0 (0) | 5 (3.2) |
| Canal | 0 (0) | 0 (0) | 3 (7.1) | 0 (0) | 3 (1.9) |
| Total water sources | 39 (100) | 40 (100) | 42 (100) | 37 (100) | 158 (100) |

* Those who can memorise the Holy Quran and can recite any time from their memory.

### 3.2. Water Source, Perceptions Related to Water Source, and Water Quality

Shallow tubewells (<76 m depth) were the most common (38%) drinking water source across the study sites. Shallow tubewell was common in Camp 4 (85%) and Camp 18 (55%) (Table 1). However, respondents did not always perceive them as safe for drinking in contrast to water from deep tubewells, which was considered safe by approximately 42% of respondents (Table 2) and which would not require treatment. They perceived water from a deep source to be tastier than other sources. Respondents preferred water from tubewells that were over 183 m deep approximately, which they considered to be safe. One female respondent said, "*There is a deep tubewell in our block, and we collect drinking water from that tubewell. The water of that tubewell is better as water comes from very deep in the ground.*" Visible dirt and insects were related to their perception of good or bad water. For example, in Camp 22, one respondent explained that, "*Supplied water is not good in quality as we found small insects comes with water.*" Similarly, in Camp 26, with the chlorinated supply, one adolescent boy commented, "*Our drinking water is a bit sour, and I feel this water is somehow contaminated as we have seen debris in drinking water*".

**Table 2.** Respondents' perceptions of clean and safe water by camp.

| Indicators | Household Point of Use Water Treatment Using Chlorine Tablets | | Piped Water Chlorination | | Overall |
|---|---|---|---|---|---|
| | Camp 4 | Camp 18 | Camp 22 | Camp 26 | |
| | N = 53n (%) | N = 57n (%) | N = 58n (%) | N = 62n (%) | N = 230n (%) |
| Desired water characteristics: perception about clean water (multiple responses allowed): | | | | | |
| Looked clear | 10 (19) | 20 (35) | 15 (26) | 23 (37) | 68 (30) |
| Absence of iron | 15 (28) | 16 (28) | 18 (31) | 19 (31) | 68 (30) |
| Absence of small insects | 18 (34) | 17 (30) | 13 (22) | 12 (19) | 60 (26) |
| No bad smell | 15 (28) | 19 (33) | 12 (21) | 14 (23) | 60 (26) |
| No bad taste | 9 (17) | 12 (21) | 11 (19) | 14 (23) | 46 (20) |
| Desired water characteristics: perception about safe water (multiple responses allowed): | | | | | |
| Looked clear | 12 (23) | 10 (18) | 10 (17) | 11 (18) | 43 (19) |
| Absence of iron | 21 (40) | 20 (35) | 19 (33) | 18 (29) | 78 (34) |
| Absence of small insects | 18 (34) | 19 (33) | 11 (19) | 10 (16) | 58 (25) |
| No bad smell | 10 (19) | 16 (28) | 13 (22) | 13 (21) | 52 (23) |
| No bad taste | 11 (21) | 9 (16) | 8 (14) | 9 (15) | 37 (16) |
| Deep tubewell (>182 m) | 25 (47) | 20 (35) | 19 (33) | 18 (29) | 82 (36) |
| Any underground source | 15 (28) | 16 (28) | 14 (24) | 9 (15) | 54 (23) |

Although study participants did not perceive alternative drinking water sources including dug wells, small streams, and canals as safe, they were used sometimes, particularly when water from regular sources was scarce during summer.

The communities with piped water chlorination systems perceived that the source water was poor, but that chlorination may improve water quality to make it safe for health. Poor quality was evident as visible dirt and debris reported by respondents. Among participants from Camp 4 and 18 who received chlorine tablets for POU treatment, 22 (out of 110) respondents were non-doers and reported that most of them used the tubewell water adjacent to their households. Respondents reported that water from sources in Camps 4 and 18 was not clean and contained iron and dirt, but had no bad smell.

Some of the respondents (30%, 68 out of 230) considered the water as clean if it looked clear and was free from visible iron (30%) and small insects (26%). Only 20% of respondents considered the absence of bad taste as a characteristic of water quality (Table 2).

### 3.3. Barriers to Water Treatment

### 3.3.1. Water Collection and Household Storage

Overall, 164/230 (71%) participants used treated water; 80% used chlorine tablets, and 63% collected water from piped water chlorination systems (Table 3). Chlorine tablets were common in Camp 4 (81%) and Camp 18 (79%), and piped water chlorination was common in Camp 22 (65%) and Camp 26 (58%).

**Table 3.** Barriers and motivators for water treatment by camp.

| Indicators | Camp 4 | | Camp 18 | | Camp 22 | | Camp 26 | | Overall | |
|---|---|---|---|---|---|---|---|---|---|---|
| | Doers | Non-Doers | Doers | Non-Doers | Doers | Non-Doers | Doers | Non-Doers | Doers | Non-Doers |
| | N = 43 n (%) | N = 10 n (%) | N = 45 n (%) | N = 12 n (%) | N = 38 n (%) | N = 20 n (%) | N = 38 n (%) | N = 24 n (%) | N = 164 n (%) | N = 66 n (%) |
| Barriers (multiple responses allowed): | | | | | | | | | | |
| Smells bad | 16 (37) | 10 (100) | 15 (33) | 12 (100) | 16 (42) | 18 (90) | 19 (50) | 21 (88) | 66 (40) | 61 (92) |
| Different taste | 16 (37) | 8 (80) | 12 (27) | 10 (83) | 16 (42) | 20 (100) | 16 (42) | 20 (83) | 60 (37) | 58 (88) |
| Inadequate supply | 8 (19) | 7 (70) | 0 | 12 (100) | 3 (8) | 7 (35) | 16 (42) | 18 (75) | 27 (16) | 44 (67) |
| Not habituated | 9 (21) | 8 (80) | 9 (20) | 11 (92) | 8 (21) | 15 (75) | 8 (21) | 17 (71) | 34 (21) | 51 (77) |
| Hair became sticky | 18 (19) | 10 (100) | 18 (40) | 10 (83) | 9 (24) | 16 (80) | 9 (24) | 18 (75) | 54 (33) | 54 (82) |
| Tablet is alive | 7 (16) | 10 (100) | 6 (13) | 10 (83) | 0 | 0 | 0 | 0 | 13 (8) | 20 (30) |
| Tattoo will be visible | 8 (19) | 9 (90) | 5 (11) | 9 (75) | 0 | 0 | 0 | 0 | 13 (8) | 18 (27) |
| Will become Christian | 6 (14) | 7 (70) | 6 (13) | 11 (92) | 0 | 3 (15) | 0 | 5 (21) | 12 (7) | 26 (39) |
| May causes death | 2 (5) | 9 (90) | 1 (2) | 9 (75) | 0 | 1 (5) | 0 | 1 (4) | 3 (2) | 20 (30) |
| Motivators (multiple responses allowed): | | | | | | | | | | |
| Kills insect/germs | 36 (84) | 3 (30) | 39 (87) | 5 (42) | 31 (82) | 15 (75) | 32 (84) | 12 (50) | 138 (84) | 35 (53) |
| Good for health | 25 (58) | 5 (50) | 31 (69) | 7 (58) | 36 (95) | 12 (60) | 35 (92) | 9 (38) | 127 (77) | 33 (50) |
| Cleans water | 23 (53) | 3 (30) | 21 (47) | 4 (33) | 33 (87) | 9 (45) | 33 (87) | 7 (29) | 110 (67) | 23 (35) |
| Removes iron | 16 (37) | 2 (20) | 18 (40) | 7 (58) | 27 (71) | 8 (40) | 12 (32) | 9 (38) | 73 (45) | 26 (39) |
| Prevents diseases | 29 (67) | 2 (20) | 28 (62) | 8 (67) | 26 (68) | 7 (35) | 29 (76) | 8 (33) | 112 (68) | 25 (38) |

Collecting water was reported as additionally burdensome in Camp 22 and Camp 26 due to the long wait times in queues (Figure 2) from shared tap-stands, since the water was only supplied during a particular time of the day, reported to be approximately half an hour on average. However, our observation revealed that almost all of the drinking water sources were located near households, within about a 5 min walk.

Observation showed that almost half of the study households stored water in containers covered with a lid. Very few (6 out of 103 households) cleaned the containers with soap. Respondents mentioned that they do not have soap or detergent to wash their utensils. One participant from a group discussion said, "*We do not have soap to bathe, how would we clean our utensils or water collection pot with soap?*".

Almost all respondents reported that women and children predominantly collected water (Figure 3). Most of the respondents (97 out of 103 households) used plastic jerrycans, aluminium pitchers, and plastic buckets to collect and store water (Figure 4).

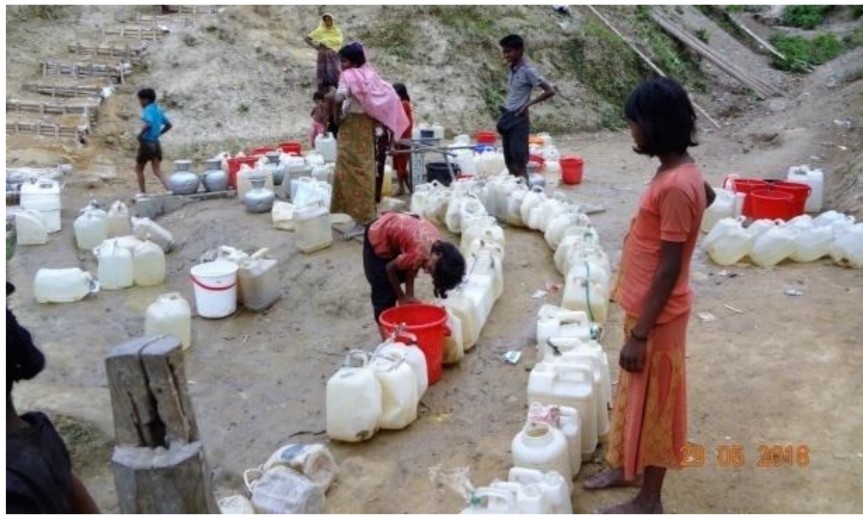

**Figure 2.** Queue for drinking water collection in Camp 26.

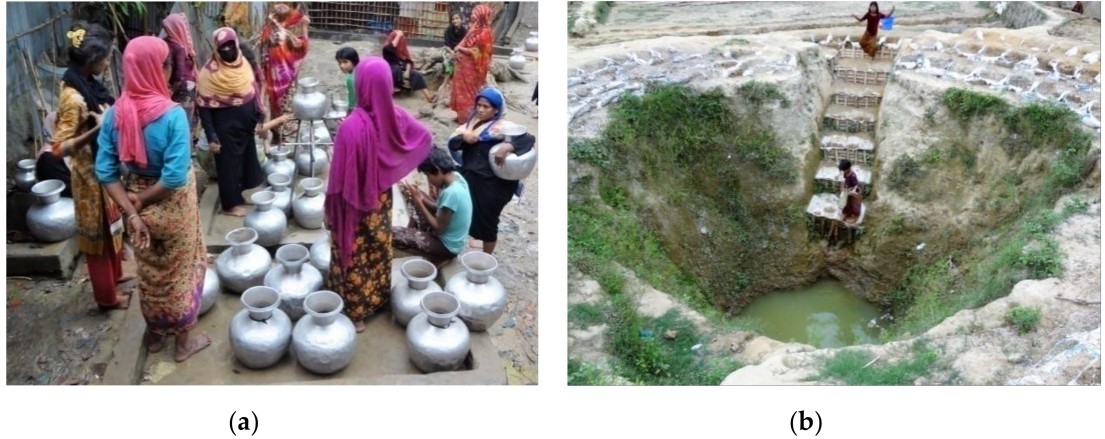

(**a**)                      (**b**)

**Figure 3.** (**a**) Water collection by women and children in Camp 4; (**b**) Water collection by children in Camp 22.

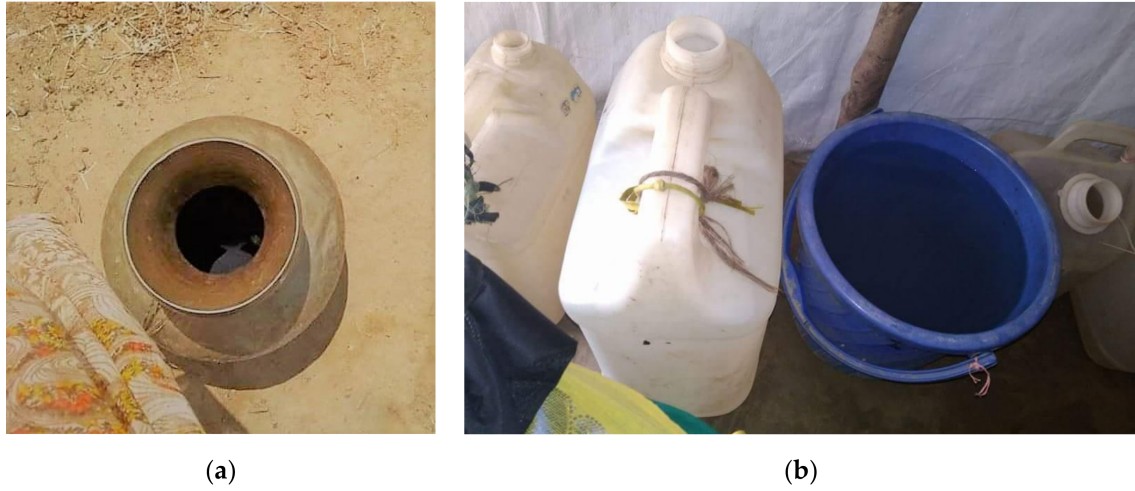

(**a**)                      (**b**)

**Figure 4.** Water collection and storage container (**a**) *Kolshi*, (**b**) jerrycan, and bucket.

### 3.3.2. Water Treatment Program Delivery and Communication with Community Members

Community meetings were arranged by volunteers and *Majhis*, who were from the Rohingya community. They received training on how to arrange and conduct meetings and what to describe during the meetings. Respondents expressed that community meetings of short duration were insufficient to understand recommendations on water treatment and use and to provide feedback. Many meetings were arranged with all community members, rather than holding separate meetings according to age, gender, or other social determinants.

Some adult male respondents mentioned that there was a need to understand the details of the water network—including some technical details regarding water supply from the large water tanks, through the pipe network, and to the tap stands that they access. Adult men and adolescent boys expressed interest in gaining information on the operation and maintenance of the system, and in particular, how damage/challenges to the tap stands could be reported. Adult women and adolescent girls reported that they had no prior experience of using tap stands and, therefore, wanted to know what differentiates these from tube wells, the common water source in the camp.

Respondents (n = 57) from Camp 18 reported that they received a bucket and sufficient chlorine tablets for two weeks and a bar of soap from NGOs, which encouraged them to treat their water.

Consistency of information about chlorine tablet distribution was an important factor in building trust in chlorination as a beneficial practice; respondents in Camp 4 and Camp 18 perceived that information on water treatment options delivered by volunteers was incorrect, since it was not consistent among all organisations, causing confusion. Respondents from Camp 18 mentioned that they received information on water treatment with an image of the treatment processes, but the volunteers also mentioned that if someone drinks the treated water just after the treatment, they would die.

### 3.3.3. Smell and Taste of Treated Water

Piped water chlorination was available in two camps, namely Camp 22 and Camp 26. The main barrier to piped water chlorination was the strong chlorine smell (42% in Camp 22; 50% in Camp 26) and taste (42% in Camp 22; 42% in Camp 26), suggesting that while they stored treated water, the taste was a characteristic of continued concern (Table 3). Water from the piped water chlorination network in Camp 22 did not meet the recommended levels of free and total chlorine for both the source (0.02 mg/L free chlorine; 0.02 mg/L total chlorine) and stored water (0.14 mg/L free chlorine; 1.5 mg/L total chlorine) (Table 4).

Chlorine tablets were available in two camps, namely Camp 4 and Camp 18. The main barrier to water treatment with chlorine tablets was the strong chlorine smell (49% in Camp 4; 47% in Camp 18) and taste (40% in Camp 4; 43% in Camp 18) (Table 3). Water from the Camp 4 and 18 did not meet the recommended levels of free and total chlorine for both the source (0.0 mg/L) and stored water (0.0 mg/L free chlorine; 0.0 mg/L total chlorine) (Table 4).

For non-doers, the most common barriers were unpleasant taste (88%) and chlorine odour (92%). Compared to non-doers, doers did not seem to notice the issue of bad smell and different taste as much, although many non-doers feared that the ingestion of the tablets might cause death (Table 3).

**Table 4.** Water residual chlorine and turbidity among Rohingya refugee camps in Cox's Bazar, Bangladesh, 2018.

| Indicators | Camp 4 | | Camp 18 | | Camp 22 | | Camp 26 | | All Samples | |
|---|---|---|---|---|---|---|---|---|---|---|
| | Source (N = 6) | Stored Water (N = 26) | Source (N = 8) | Stored Water (N = 23) | Source (N = 7) | Stored Water (N = 34) | Source (N = 5) | Stored Water (N = 20) | Source (N = 26) | Stored Water (N = 103) |
| | n (%) | n (%) | n (%) | n (%) | n (%) | n (%) | n (%) | n (%) | n (%) | n (%) |
| *Turbidity (NTU):* | | | | | | | | | | |
| <5 | 5 (83) | 17 (65) | 1 (13) | 5 (22) | 2 (29) | 13 (38) | 4 (80) | 18 (90) | 11 (42) | 62 (51) |
| Mean NTU (SD) | 5 (7.9) | 6.3 (8.2) | 1 (3.4) | 3 (2.9) | 14 (12) | 11.7 (11) | 5.5 (11) | 0.4 (2.9) | 5.1 (10) | 7.1 (10) |
| *Free chlorine (mg/L):* | | | | | | | | | | |
| <0.2 | 5 (83) | 26 (100) | 8 (100) | 19 (83) | 6 (86) | 28 (82) | 4 (80) | 19 (95) | 23 (88) | 111 (91) |
| 0.2–2 | 0 | 0 | 0 | 1 (4.4) | 0 | 6 (18) | 0 | 1 (5.0) | 0 | 8 (6.6) |
| Mean mg/L (SD) | 0.04 (0.04) | 0.05 (0.03) | 0.04 (0.02) | 0.59 (0.55) | 0.02 (0.01) | 0.14 (0.21) | 0.02 (0.02) | 0.08 (0.10) | 0.03 (0.03) | 0.09 (0.07) |
| *Total chlorine (mg/L):* | | | | | | | | | | |
| <0.2 | 5 (83) | 26 (100) | 8 (100) | 19 (83) | 6 (86) | 23 (68) | 4 (80) | 12 (60) | 23 (88) | 99 (81) |
| 0.2–2 | 0 | 0 | 0 | 1 (4.4) | 0 | 2 (5.9) | 0 | 8 (40) | 0 | 11 (9.0) |
| >2 | 1 (17) | 0 | 0 | 3 (13) | 1 (14) | 9 (26) | 1 (20) | 0 | 3 (11) | 12 (9.8) |
| Mean mg/L (SD) | 0.03 (0.05) | 0.02 (0.02) | 0.03 (0.02) | 0.61 (0.68) | 0.02 (0.01) | 1.5 (1.3) | 0.03 (0.03) | 0.21 (0.24) | 0.03 (0.03) | 0.57 (0.53) |

### 3.3.4. Fear of Religious Conversion

We detected many examples of strong negative perceptions and rumours associated with water treatment using chlorine tablets. Twelve respondents (out of 164 doers) and 26 respondents out of 66 non-doers reported that they were concerned that if they used chlorine tablets, then tattoos might appear on their bodies and perceived it as a mark of Christian faith. One respondent said, "*If we drink that chemical mixed water, we will not remain Muslim and will be converted to Christianity; tattoos will be visible on our arms once we become Christian. We have seen the same tattoo on the arms of Christian foreigners*".

### 3.3.5. Inadequate Supply of Water Treatment Products

Some of the chlorine tablet users (17 out of 22) reported that the number of tablets they received was not adequate and that there were often delays in replenishing the supplies. Another barrier was the limited number of containers in households to treat water with tablets and wait for the prescribed period before drinking. A large number of doers reported an inability to develop a regular water treatment habit (21%) as another barrier for drinking water treated with chlorine tablets, in addition to the inadequate supply of tablets (Table 3).

### 3.4. Motivators for Water Treatment

### 3.4.1. Perceived Health Benefits

The respondents explained that the most common motivating factor for the uptake of any chlorination method was the perceived health benefits (Table 3). Among those who were drinking water from the piped water chlorination system, 68% reported that fewer had suffered from diseases like dysentery and diarrhoea than prior to installation of the system. Respondents insisted that children should drink treated water to save them from diseases since, given their living conditions, it would be difficult to manage and take care of sick children.

The perception that chlorination "killed germs/insects|" was commonly reported by "doers" (138 out of 168) but less commonly by "non-doers" (35 out of 66), who used shallow tubewell water (Table 3). Doers (67%) more commonly reported the chlorination "cleaned water" by reducing iron and dirt, but "non-doers" less commonly reported this aspect (35%).

### 3.4.2. Ease of Use

Respondents using both chlorination methods found them easy to use. Those collecting water from the piped water chlorination method did not need to do anything except collect and store water. Those using chlorine tablets stated that they only had to drop a tablet into their water container (jerrycans with 10 L capacity provided with chlorine tablets).

### 3.5. Residual Chlorine in the Water

Among collected samples, there were low levels of turbidity in most of the source water (mean ≤ 5 NTU), except in Camp 22, where households used water from the piped water chlorination system (mean = 14 NTU, SD =12.0). Stored water turbidity was similarly low in all camps except in Camp 22 (mean = 11.7 NTU, SD = 11). Among household stored water samples, 6.6% met the WHO recommendation (0.2–2 mg/L) for free residual chlorine levels (mean = 0.09 mg/L (SD 0.07)). Ninety-one per cent of household stored water had <0.2 mg/L free residual chlorine (Table 4). However, we did not find any free chlorine in the source water of Camp 22.

## 4. Discussion

Our study confirmed findings from other settings that a common barrier to chlorinated water uptake is the smell or taste of treated water [11,14,15,18,19], even in the emergency context. Chlorine-based water treatment faces barriers to acceptability, as reported in numerous field trials of

household water chlorination products [20–22]. While the Rohingya refugees resided in Myanmar, they consumed untreated groundwater (tubewell water). The concern about smell and taste are challenging to address within a short period among communities with no prior exposure to chlorinated water; Rohingya refugees would be more likely to continue their habit of drinking untreated tubewell water. Our findings indicate that the introduction of the water treatment program without consideration of the socio-cultural aspects of the community is unlikely to be successful or sustainable. However, our study indicates that the humanitarian context is different from a relatively stable environment in many ways, and is therefore quite challenging for the initial phases of the water treatment programs [23]. Additionally, this study found that a continuous and uninterrupted supply of chlorine tablets or piped water chlorination was necessary for community habit formation [15].

Smell was the predominant barrier to using chlorine products; however, this varied among doers versus non-doers, where doers were less likely to complain about smell compared to non-doers. A study on chlorine detection and acceptability thresholds for water treatment in Dhaka found that the acceptability limit (1.25 mg/L) was well under the maximum free chlorine target dose (2.0 mg/L), which clearly indicates that users can only accept 60% of the maximum limit of free chlorine [12]. As chlorine acceptance thresholds likely vary by region, culture, and water quality, it is important to conduct a chlorine acceptance study prior to initiating a chlorination program. Moreover, when introducing water treatment to communities, clearly stating that taste and smell are typical characteristics of chlorine-treated water may help to improve uptake, especially among communities where chlorination has not been used. Additionally, reassuring the community that high-income countries have used chlorine for decades to provide safe water to their communities may allay the fears of possible hazardous chemicals being used. In future programs, managers should consider determining the optimal chlorine dose through small scale experiments, as water chemistry impacts chlorine demand [24] and smell/taste. Noting that water treatment can result in water with a clean, clear appearance could be seen as an advantage and thereby increase uptake, since this seems to be a characteristic that the community values.

De-motivating factors related to trustworthiness of the water sources for chlorination, water quality, and fears of adverse effects of treating water with "chemicals/medicine" should be part of introductory discussions about the interventions. Water treatment must be rolled out to maximise trust and early uptake to best set up habit formation. Community misconceptions are significant barriers to public health programs [25]. Beliefs, practices, and cultural norms overshadow public health priorities and ethics [26]. False beliefs regarding services and distrust of healthcare workers or frontline workers can disrupt the success of interventions [27]. A lack of community trust can render programs unsuccessful. In this study, participants expressed concerns related to their religious beliefs. In a study in Pakhtun, Pakistan among health journalists, they found that weak community trust in the government, security concerns, and community members' religious beliefs were major impediments to increasing the uptake of the polio vaccine [27,28]. Human behaviour, including the utilisation and acceptability of healthcare services, is greatly influenced by religious beliefs and dogma [26]. We also detected issues of suspicion and low-level trust, which should be expected among a community persecuted on religious grounds. Addressing religious concerns like fear of religious conversion is important among Rohingyas, whose religion was the primary basis of their discrimination and subsequent refuge in Bangladesh. Involving religious leaders to lead communications by possibly reciting the Quran before commencing and/or using mosques as a delivery platform for accurate messages, in their language, should be considered. Implementing chlorine-based water treatment at mosques could potentially set a community norm.

Public health programming often adopts an instructive approach rather than building on the local expertise of affected communities. However, the simple provision of hardware and facilities does not necessarily ensure that crisis-affected populations will use them effectively. The reasons for this are complex social norms, perceptions of risk, and the availability of resources, which can all influence whether certain positive changes are adopted. In addition, program managers may not measure whether

intervention delivery results in meaningful participation. Community members should be included in identifying local priorities, problems, and their own solutions. In addition, specific gendered needs of women and men and boys and girls should be taken into account in the design and location of the facilities. Using a mix of communication channels and community mobilisation approaches could increase program uptake. It is important to facilitate community dialogue and action about water treatment in various groups and organisations to improve skills, clarify misconceptions, and keep the practice in the minds and lives of individuals and families up to the point when water treatment practices become a household habit and a new social norm.

A quick survey of the users is required prior to implementing the intervention, which could support the estimation of supplies and hardware for the community. The implementing organisation needs to develop a close monitoring system for water treatment with chlorine methodologies to define chlorine doses better from the beginning. Even though shallow tubewells were the most common source of drinking water, access to the sources was challenging. Visits to collect water for the household were frequent, due to insufficient capacity and/or the number of storage containers to meet the daily needs of the whole family, which future programs should take into account. Where finances permit, arranging or distributing containers might be an appropriate solution. A previous study [19] showed that refugees in Liberian camps had to collect water several times a day due to lack of storage containers, which encouraged drinking untreated water, even though there were chlorine tablets in the home. In future interventions, safe storage could be promoted [29].

Household-level water chlorination methods require individuals to properly and consistently treat their own water to realise health benefits [30,31]. Though impacts of WASH interventions on health outcomes in humanitarian settings is limited [32], promoting health benefits of water treatment, especially among non-doers, is crucial for sustained adoption. Field studies by Rosa and colleagues in India, Zambia, and Peru suggest exaggerated consistency of practices and suboptimal microbiological effectiveness, even among householders that report usually treating their water at home before drinking [33]. Chlorine treatment based on WHO guidelines for drinking water quality may not ensure that water remains safe over its entire course in the setting of a refugee camp [34]. Therefore, promoting health benefits as well as safe water collection and storage is crucial to achieve "safe water for all".

### 4.1. Limitations

Although qualitative research is a common approach to explore existing practices and contexts within a community, we conducted this study in four camps, and therefore caution should be used in generalising the findings across the Rohingya camps. To maximise variability, we included a variety of camps in the study to cover a broad context. Thus, the findings provide useful insights into the limitations of water treatment initiatives as well as data to support improvements to the current programs implemented in camp areas. We did not conduct laboratory procedures for the presence of harmful microorganisms in drinking water. However, other studies were ongoing simultaneously that conducted laboratory procedures and found approximately 28% of the source water was contaminated with faecal coliforms and 10.5% with *E. coli* among 3186 tubewells tested [1,8].

### 4.2. Recommendations

When rolling out a chlorine-based intervention, it seems prudent to determine whether the end users have used chlorinated water previously, especially for drinking. The predominant barriers to chlorine uptake were the smell and taste, likely due to no prior exposure to chlorinated water. These two issues can often occur with inaccurate chlorine dosing or by misunderstanding the proper use of household chlorination tablets. Including instructions on how to minimise smell and taste (e.g., by allowing water to stand for additional time before drinking, using Aquatabs at the point of collection rather than dosing at the household, and dosing the night before the following day's

supply) via communications and undertaking "taste tests" with communities could also help to identify acceptable levels of chlorine within recommended standards.

Working with community influencers (such as *Imams*, *Hafez*, and *Hafeza*, community volunteers) in interactive question and answer sessions to demonstrate how water treatment works, or how chlorine solutions are made, or even demonstrations of the application of Aquatabs can be used to address concerns. Religious leaders could lead communications in the preferred language. They could deliver the messages in mosques in order to implement chlorine-based water treatment for habituating it into usual practice.

Providing information on impacts of contaminated water on health and explaining pathways of contamination could be added in future interventions, as the health benefits of drinking treated water were found as a motivator in this community.

Furthermore, substantial efforts are required in terms of preparedness and potential crisis-mapping based on epidemiological data, socio-economic factors, and the historical past. These efforts will be relevant to understand the potential scale and scope of new crises, especially in protracted contexts.

**Author Contributions:** Data curation, S.F.; Formal analysis, S.M.M.A.; Supervision, A.A. and S.S.; Writing—original draft, M.-U.A.; Writing—review and editing, L.U., N.A., D.B., and M.R. All authors have read and agreed to the published version of the manuscript.

**Funding:** The study has been funded by Oxfam.

**Acknowledgments:** icddr,b acknowledges with gratitude the commitment of OXFAM. icddr,b is thankful to the Governments of Bangladesh, Canada, Sweden, and the UK for providing core/unrestricted support. We are also thankful to Daniele Lantagne, Eva Niederberger, and Michelle Farington for their support in study design and study report preparation.

**Conflicts of Interest:** The authors declare no conflict of interest.

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
