# Peer review of "Barriers and Enabling Factors for Central and Household Level Water Treatment in a Refugee Setting: A Mixed-Method Study among Rohingyas in Cox’s Bazar, Bangladesh"

_water, doi:10.3390/w12113149_

Round 1

Reviewer 1 Report

This is an interesting paper and easy to read, except for tables. Good to have recommendation in the end build on lessons learned in the project for future work. Some important lessons from the research could be added to recommendation as e.g. provide information on impact of contaminated water on health and pathways of contamination (68% reported that fewer had suffered from waterborne diseases page 11 line 304-305)

My main comments are that tables are unclear and confusing and I would suggest reviewing and improving them as possible to clarify 

  • Percentage to not add up (probably because interviewee can chosen more than one answers). 
  • In Table 4 is expecially confusing.
    • Turbidity zero in Camp 18, or was it not measured?
    • N=7 in Camp 22 but 13 samples <5 NTU in stored water. 
    • Confusing to have in brackets either % or SD in the same table. 

Author Response

This is an interesting paper and easy to read, except for tables. Good to have recommendation in the end build on lessons learned in the project for future work. Some important lessons from the research could be added to recommendation as e.g. provide information on impact of contaminated water on health and pathways of contamination (68% reported that fewer had suffered from waterborne diseases page 11 line 304-305)

Response: Thanks for your suggestions. We have added below sentence in the recommendations (Line 452-454):

“Providing information on impact of contaminated water on health and explaining pathways of contamination could be added in the future intervention as health benefits of drinking treated water was found as a motivator in this community.”

My main comments are that tables are unclear and confusing and I would suggest reviewing and improving them as possible to clarify 

  • Percentage to not add up (probably because interviewee can chosen more than one answers).

Response: Thanks for this detailed observation. We have now further explained that multiple responses were allowed to improve clarity. We have added a note beside the indicator, in the table 2 and 3.

  • In Table 4 is especially confusing.

Response: Thanks for your suggestions. we have revised the table that improves clarity.

  • Turbidity zero in Camp 18, or was it not measured?

Response: Thank you so much; it was an error, we have corrected the information (Table 4).

  • N=7 in Camp 22 but 13 samples <5 NTU in stored water.

Response: Thanks for comment. Now, source water N and stored water N is included in the table heading to avoid confusion.

  • Confusing to have in brackets either % or SD in the same table. 

Response: Thanks for this suggestions. We have added n (%) at the table sub-heading.

Reviewer 2 Report

The health risks associated with water consumption are very serious. Therefore, many studies related to assessment of this risk are conducted around the world. Lack of treated water increases the risk of illness and/or death.

 However, the presented results of research are mainly of a sociological nature.

There is no statistical assessment of the risk of treated and untreated water consumption.  

The paper mentions high doses of chlorine. Therefore, it should also analyze the concentration of disinfection by-products.

There is also no information on the quality of water from individual sources.

Author Response

The health risks associated with water consumption are very serious. Therefore, many studies related to assessment of this risk are conducted around the world. Lack of treated water increases the risk of illness and/or death.

However, the presented results of research are mainly of a sociological nature.

There is no statistical assessment of the risk of treated and untreated water consumption.  

The paper mentions high doses of chlorine. Therefore, it should also analyze the concentration of disinfection by-products.

There is also no information on the quality of water from individual sources.

Response: Thank you so much for your comments. Yes, this is mainly a qualitative study, as we have pointed out in the title, key words, methods sections. The qualitative components were designed to collect data in a way that provides respondents a voice, collecting opinion and open ended comments that are not captured during quantitative surveys. In this instance we needed to determine why water treatment interventions may be failing in a population that differs from the broader community, in order to inform changes that are more likely to impact intervention uptake.

In qualitative studies, data are analysed thematically and therefore there is no scope to use statistical methods. Instead, we have explored the perception of the participants about the risk of treated and untreated water consumption.

The study was not designed as an impact evaluation and therefore data were not collected at baseline and after intervention, from intervention and control communities.

We have collected treated water to see residual chlorine both from piped water supply and chlorine tablet distribution area. We have showed the chlorine concentration in those area, which actually indicates by-products also.

We didn’t collect water quality information from individual sources except turbidity, shown in table 4.

We hope that above clarification satisfies your queries.

Reviewer 3 Report

This study was aimed to present field-derived end-user perspectives on water quality and accessibility in a refugee setting. Authors are commended for their efforts in this important, but under-researched theme that is often neglected in published literature. To this end, the manuscript has the potential to make a good contribution to the sector (not just academics, but also practitioners). Overall, there are no major issues with the presentation of the work. However, some deficiencies have been identified and if addressed could bolster the quality of the presented study, namely:

  • Methods: Camps 22 (Unchiprang) and 26 (Nayapara) have a water supply system that is supplied by a surface water treatment plant. This is not clear from the methods description that refers to these as "chlorinated" water supplies. The manuscript narrative is one that omits this and implies that the only treatment in these camps is chlorination. This is not accurate. Its believed that the other camps derive their water from ground water sources, but this is also not clear. A better description of the water supply sources is warranted.
  • Methods: It seems the authors have collected information from human subjects. However, I was surprised not to see any mention of an ethics review, informed consent, and anonymised information collection.
  • Methods: Later in the manuscript (for example line 212), it is apparent that visible water quality ("dirt and debris") were used as part of the assessment. This was not included in the Methods section. It is also not clear if this is based on a direct observation by measurers or self-reported information by participants.
  • Methods: Mention of iron levels is made (e.g. line 217), but Methods lack description of how this parameter was assessed.
  • Results: Table 1 indicates that 0 % of participants used deep tubewells in Camps 22 and 26. However, Table 2 indicates that 33 and 29 % of participants in those sites indicated desired characteristics for that type of water source. It is not clear how participants expressed a perception of a water source they did not use. It seems that a quality check of the data may be needed or at the very least an explanation of such a discrepancy.
  • Results: Line 272 does not specify what type of residual is reported, free or total? Also, is there a mistake? Recommended level is 0.2 mg/L and not 0.02 mg/L as reported.
  • Discussion: Authors should revise the use of subjective language such as "...barriers are difficult to overcome..." (line 330). Please qualify the difficulty.
  • Discussion: Lines 332-334 suggests that water supply was made entirely devoid of any consideration of socio-cultural characteristics of the community. I can appreciate that in hindsight there could have been scope for improvement in how things were done. However, the manuscript gives the reader little appreciation for the dynamic situation in which these water supply systems were set up. Also, there is little in terms of recommendation of how it could have been done better.
  • Discussion: Lines 348-352 are lacking references to support such statements.
  • Discussion: In line with the previous comment, I was surprised to note that authors did not engage with recent relevant literature on the theme of this paper: doi.org/10.1016/j.watres.2020.115854, doi.org/10.3390/w11061309, doi.org/10.3390/w12051506

Author Response

This study was aimed to present field-derived end-user perspectives on water quality and accessibility in a refugee setting. Authors are commended for their efforts in this important, but under-researched theme that is often neglected in published literature. To this end, the manuscript has the potential to make a good contribution to the sector (not just academics, but also practitioners). Overall, there are no major issues with the presentation of the work. However, some deficiencies have been identified and if addressed could bolster the quality of the presented study, namely:

  • Methods: Camps 22 (Unchiprang) and 26 (Nayapara) have a water supply system that is supplied by a surface water treatment plant. This is not clear from the methods description that refers to these as "chlorinated" water supplies. The manuscript narrative is one that omits this and implies that the only treatment in these camps is chlorination. This is not accurate. Its believed that the other camps derive their water from ground water sources, but this is also not clear. A better description of the water supply sources is warranted.

Response: Thanks for this suggestions, we have added the following text in Line 98-100:

“In Camp 22 and Camp 26, water from a surface water treatment plant was provided from an established supply network. In Camp 4 and Camp 18, inhabitants used various water sources including shallow tube wells, deep tubewells or a small stream.”

  • Methods: It seems the authors have collected information from human subjects. However, I was surprised not to see any mention of an ethics review, informed consent, and anonymised information collection.

Response: Thanks for indicating this unintentional error. We have added ethics information in the revised manuscript (Line 186-193). We have added following text:

“2.6. Ethical Approval

The study protocol was approved by the Ethics Review Committee of the International Centre for Diarrhoeal Disease Research, Bangladesh (icddr,b). Study participants were informed of the aims of the study and their rights. Enumerators read an information sheet to respondents in local language, answered any questions raised, and obtained written consent for participation. Respondents were given a copy of the information sheet to keep, and no compensation was provided for participation. Names and numbers were removed from final data sets to protect anonymity.”

  • Methods: Later in the manuscript (for example line 212), it is apparent that visible water quality ("dirt and debris") were used as part of the assessment. This was not included in the Methods section. It is also not clear if this is based on a direct observation by measurers or self-reported information by participants.

Responses: This was self-reported information by participants. We have added in the methods as “perceptions related to water quality” in line 135. We have edited results also as “Poor quality was evident as visible dirt and debris reported by respondents” (Line 224).

  • Methods: Mention of iron levels is made (e.g. line 217), but Methods lack description of how this parameter was assessed.

Response: This was self-reported data by participants. We have added the word “visible” in line 230 to clarify this.

  • Results: Table 1 indicates that 0 % of participants used deep tubewells in Camps 22 and 26. However, Table 2 indicates that 33 and 29 % of participants in those sites indicated desired characteristics for that type of water source. It is not clear how participants expressed a perception of a water source they did not use. It seems that a quality check of the data may be needed or at the very least an explanation of such a discrepancy.

Response: Thanks for this comments. We have checked all data and made necessary edits. Those data were correct as during the study period they were not using deep tube well water, but they have a perception that deep tube well water is safe as they used similar sources in Myanmar, before escaping to Bangladesh. Table 1 is about their current water source, and Table 2 is about their desired water source, so it is possible that there could be a difference between these two tables.

  • Results: Line 272 does not specify what type of residual is reported, free or total? Also, is there a mistake? Recommended level is 0.2 mg/L and not 0.02 mg/L as reported.

Response: Thanks for indicating that error. We have edited the sentence. Now the text reads-

“Though water from the piped water chlorination network in Camp 22 did not meet the recommended levels of free and total chlorine for both the source (0.02 mg/L free chlorine; 0.02 mg/L total chlorine) and stored water (0.14mg/L free chlorine; 1.5 mg/L total chlorine).”

  • Discussion: Authors should revise the use of subjective language such as "...barriers are difficult to overcome..." (line 330). Please qualify the difficulty.

Response: Thanks for this suggestions. We have revised the sentence:

“The concern about smell and taste are challenging to address within a short period…”

  • Discussion: Lines 332-334 suggests that water supply was made entirely devoid of any consideration of socio-cultural characteristics of the community. I can appreciate that in hindsight there could have been scope for improvement in how things were done. However, the manuscript gives the reader little appreciation for the dynamic situation in which these water supply systems were set up. Also, there is little in terms of recommendation of how it could have been done better.

Response: We have added one sentence (Line 352-353):

“However, our study indicates that the humanitarian context is different from a relatively stable environment in many ways, therefore quite challenging for the initial phases of the water treatment programs [24].”

To clarify further, we have already talked about how it could be better (Line 444):

“Undertaking ‘taste tests’ with communities could also help to identify acceptable levels of chlorine within recommended standards.”

  • Discussion: Lines 348-352 are lacking references to support such statements.

Response: We have added appropriate references.

  • Discussion: In line with the previous comment, I was surprised to note that authors did not engage with recent relevant literature on the theme of this paper: doi.org/10.1016/j.watres.2020.115854, doi.org/10.3390/w11061309, doi.org/10.3390/w12051506

Response: Thank you so much for your comments. We have tried to connect our discussion with suggeste

Round 2

Reviewer 1 Report

Accept the improvements you have made to manuscript and suggest publication.   

Reviewer 2 Report

In my opinion stil  the presented results of research are mainly of a sociological nature.

Reviewer 3 Report

Authors have addressed the previously identified issues.